# Comparison of Benzbromarone and Allopurinol on Primary Prevention of the First Gout Flare in Asymptomatic Hyperuricemia

**DOI:** 10.3390/jpm12050697

**Published:** 2022-04-27

**Authors:** Shih-Wei Lai, Kuan-Fu Liao, Yu-Hung Kuo, Chiu-Shong Liu, Bing-Fang Hwang

**Affiliations:** 1Department of Public Health, College of Public Health, China Medical University, Taichung 404, Taiwan; wei@mail.cmuh.org.tw; 2School of Medicine, College of Medicine, China Medical University, Taichung 404, Taiwan; d3350@mail.cmuh.org.tw; 3Department of Family Medicine, China Medical University Hospital, Taichung 404, Taiwan; 4College of Medicine, Tzu Chi University, Hualien 97004, Taiwan; kuanfuliaog@gmail.com; 5Department of Internal Medicine, Division of Hepatogastroenterology, Taichung Tzu Chi Hospital, Taichung 427, Taiwan; 6Department of Research, Taichung Tzu Chi Hospital, Taichung 427, Taiwan; tc1921602@tzuchi.com.tw; 7Department of Occupational Safety and Health, College of Public Health, China Medical University, Taichung 404, Taiwan

**Keywords:** allopurinol, asymptomatic hyperuricemia, benzbromarone, cohort study, first gout flare

## Abstract

Objectives. Whether uric acid-lowering agent use in asymptomatic hyperuricemia can reduce the development of the first gout flare remains unsettled. The goal of the present research was to test the efficacy of benzbromarone and allopurinol on primary prevention of the first gout flare in persons with asymptomatic hyperuricemia in Taiwan. Methods. One observational cohort study was constructed to examine the 2001–2015 dataset adapted from the National Health Insurance Program of Taiwan containing the claims data of 2 million beneficiaries. Asymptomatic hyperuricemia was considered as individuals on uric acid-lowering therapy who did not have gout flares. Individuals aged 20–84 without gout flares who had the use of benzbromarone alone were assigned into a benzbromarone group. Individuals ages 20–84 without gout flares who had the use of allopurinol alone were assigned into an allopurinol group. The final study included 6111 pairs of 1:1 propensity score-matched individuals from both benzbromarone and allopurinol groups. The end point was assigned as individuals who were newly diagnosed with their first gout flare. The incidence rate of the first gout flare was estimated between the benzbromarone and allopurinol groups. A Cox proportional hazards regression model was applied to explore the hazard ratio and 95% confidence interval of the first gout flare related to benzbromarone use and allopurinol use. Results. The incidence rate of the first gout flare was lower in the benzbromarone group compared with an allopurinol group (3.29 versus 5.46 per 1000 person-months, incidence rate ratio = 0.60 and 95% confidence interval = 0.56–0.64). After adjustment for co-variables, the adjusted hazard ratio of the first gout flare was 0.63 (95% confidence interval = 0.59–0.68, *p* < 0.001) for the benzbromarone group when compared with the allopurinol group. Conclusion. People with asymptomatic hyperuricemia taking benzbromarone have a lower hazard of developing their first gout flare when compared with those taking allopurinol. Based on the medication safety, the therapeutic effects and the low price, with oral administration once daily, we suggest that benzbromarone should be the first drug of choice if clinicians are treating asymptomatic hyperuricemia.

## 1. Introduction

Uric acid (UA) is a major metabolic product of human purine. A high serum UA level (also known as hyperuricemia) may elevate the probability of gout flares [1,2]. Gout is regarded as an inflammatory joint condition, and it develops as the result of the deposition of monosodium urate crystal within joints and/or in soft tissues [3,4,5]. The majority of persons with hyperuricemia will not have a gout flare over a long time, and this condition is usually defined as asymptomatic hyperuricemia [1,2]. However, gout is always initiated by hyperuricemia.

Key messages
The hazard ratio of the first gout flare was 0.63 for a benzbromarone group when compared with an allopurinol group.Benzbromarone should be the first drug of choice if clinicians are treating asymptomatic hyperuricemia.


The prevalence of gout showed a geographic variation. The prevalence of gout in USA adults between 2007 to 2008 (affecting 8.3 million individuals) and in 2015–2016 (affecting 9.2 million individuals) was similar, about 3.9% [6]. The prevalence of gout among Taiwan adults by examining the 2000–2002 dataset of one medical center were 3.1% (5% in male and 0.6% in female) [7]. A recent review reported that about 7.44 million new gout cases were estimated in the world in 2017 [8]. The global incidence rate and the prevalence of gout steadily elevated during the last 25 years (from 1992–2017) [8].

Hyperuricemia is considered as the key risk factor for gout flares [3,9,10]. A systematic review has shown that the probability of gout recurrence ranged from 12% (the serum UA level ≤ 6 mg/dL) to 61% (the serum UA level ≥ 9 mg/dL) in people taking uric acid-lowering agents [11]. The higher serum UA level was related to a higher probability of gout recurrence in a graded relationship [11].

Numerous risk factors may serve as the risk for hyperuricemia, including (1) overproduction-type, such as the excess consumption of purine-rich foods, excess alcohol intake, excess consumption of fructose, hematological diseases and others; (2) renal underexcretion-type, such as aspirin (low-dose), thiazide diuretics, loop diuretics, obesity, hypertension, metabolic syndrome, chronic renal failure and others [12,13]. The general therapeutic principle of hyperuricemia is to eliminate these risk factors. However, sometimes medical treatment is still needed.

There are two types of urate-lowering agent available in the market. Xanthine oxidase inhibitors are indicated for people with overproduction-type hyperuricemia [14,15]. Uricosuric agents are indicated for people with renal underexcretion-type hyperuricemia or for those with mixed-type hyperuricemia [14,15]. Urate-lowering agents have a secondary preventive effect against the recurrence of gout flares in hyperuricemic people who already have suffered from gout flares [11]. Recently, one review showed that no definite evidence proved that allopurinol use (one xanthine oxidase inhibitor) could be associated with reduced probability of the first gout flare in asymptomatic hyperuricemia [16].

Benzbromarone (one uricosuric agent) had been withdrawn from some countries since 2003 owing to its potential hepatotoxicity [17]; however, it is still available in Taiwan. Whether benzbromarone use can reduce the probability of the first gout flare in asymptomatic hyperuricemia remains unknown. The current evidence appears to be not sufficient to evaluate the balance of benefit and harm of urate-lowering agents use in asymptomatic hyperuricemia. If we want to recommend a clinical practice, it would be important to know which agent is the most effective one that can avoid the first gout flare and can radically change the natural history of asymptomatic hyperuricemia.

Giving that gout causes major medical and economic impacts on patients and employers owing to its joint disability, work absence and subsequent decreased productivity, [18,19] a real-world database was analyzed to investigate the following issues: (1) the efficacy of benzbromarone and allopurinol on primary prevention of the first gout flare among people with asymptomatic hyperuricemia and (2) the proportion of achieving a treatment target for benzbromarone users and allopurinol users.

## 2. Methods

### 2.1. Research Design and Data Collection

This was an observational cohort research utilizing the 2001–2015 dataset adapted from the National Health Insurance Program in Taiwan (NHIP), which contained the claims data of 2 million beneficiaries. The NHIP began in Taiwan on March 1 in 1995 [20]. By the end of 2018, approximately 99.7% of the whole 23 million people living in this country had been covered by the program [20]. In brief, the reimbursement claim data of every insured people including birth date, sex, and all utilized medical services were recorded in the database of the NHIP. In order to ensure confidentiality, the people’s identification numbers were encrypted. The reimbursement claim data were available to the public.

### 2.2. Inclusion Criteria and Exclusion Criteria of Study Population

The mainly pharmacological effect of urate-lowering drugs is to lower the serum uric acid. Therefore, individuals who take any urate-lowering agent must have hyperuricemia. In this present research, asymptomatic hyperuricemia was considered as individuals who took urate-lowering agents and did not have gout flares. An index date of the research was regarded as the date of prescribing urate-lowering agents. Only those individuals with cumulative time of drug usage from the index date to the endpoint ≥ 30 days were selected for the research.

Individuals ages 20–84 who had the use of benzbromarone alone on the index date were assigned into the benzbromarone group. Individuals ages 20–84 who had the use of allopurinol alone on the index date were classified as an allopurinol group.

Individuals with the below conditions were excluded from the research: (1) individuals who suffered from gout flares before an index date, (2) individuals who experienced other gout-related disorders, including gouty tophi, gouty urolithiasis and gouty nephropathy prior to an index date and during the follow-up time, (3) individuals who experienced any cancer prior to an index date and the follow-up time, and (4) individuals who had alternate prescriptions of uric acid-lowering agents during the cohort. Figure 1 presents a flow chart of selection of study subjects.

### 2.3. Main Endpoint

The main endpoint was defined as individuals who were newly diagnosed with the first gout flare on joints (according to International Classification of Diseases 9th Revision Clinical Modification, ICD-9 codes 274.0 and 274.9). A previous study assessed the correctness of the gout diagnosis in the basis of ICD-9 codes, with a sensitivity of 78% and a specificity of 100% [21]. 

### 2.4. Baseline Medications and Comorbidities

Urate-lowering agents available in Taiwan included allopurinol, febuxostat, benzbromarone, sulfinpyrazone and probenecid. The baseline medications included thiazide diuretics, loop diuretics and aspirin, all of which could be related to hyperuricemia. The baseline comorbidities included cerebrovascular disease, chronic kidney disease (CKD), chronic obstructive pulmonary disease (COPD), coronary artery disease, diabetes mellitus (DM), hyperlipidemia and hypertension. The baseline comorbidities were diagnosed in the basis of ICD-9 codes.

### 2.5. Sup-Population Analysis

In order to evaluate the efficacy of benzbromarone use and allopurinol use, we collected the clinical data for one local hospital in Taiwan over 2017–2020. Only those individuals with cumulative time of drug usage ≥30 days were included. Only those individuals with the baseline serum UA level ≥ 6.8 mg/dL were included. The following variables were included: (1) sex and age, (2) the baseline serum UA level, (3) the serum UA level at 12 weeks of treatment, and (4) the proportion of achieving the treatment target at 12 weeks (serum UA level < 6 mg/dL).

### 2.6. Statistical Analysis

We utilized the Chi-square test to assess the categorical variables between benzbromarone users and allopurinol users. The *t*-test was utilized to assess the continuous variables. The incidence rates of the first gout flare were measured for benzbromarone users and allopurinol users stratified by sex and age groups. The cumulative incidence of the first gout flare was presented by the Kaplan–Meier curve. The difference of the cumulative incidence was assessed by using a log-rank test. A Cox proportional hazard regression model was utilized to assess a hazard ratio (HR) and 95% confidence interval (CI) for the first gout flare with association of baseline medications and comorbidities. 

The proportional hazard assumption was examined by Cumulative Sums of Martingale Residuals and a Kolmogorov-type Supremum test. It was not violated. We performed a propensity score matching model based on sex, age and comorbidities. We applied a logistic regression model to measure the propensity score for the two drug groups. Individuals in the two drug groups were matched 1:1 on the basis of the nearest-neighbor matching method without replacement by applying a caliper width equal to 0.2-times the standard deviation of the logit of the propensity score. We ran the analyses utilizing SAS software (version 9.4 for Windows; SAS Institute Inc., Cary, NC, USA). A *p* value less 0.05 was considered to be statistically significant.

## 3. Results

### 3.1. Characteristics of Information of the Study Individuals

After propensity score matching, there were 6111 individuals in the benzbromarone group and 6111 individuals in the allopurinol group (Table 1). The two groups had similar distributions in sex and age, with 65% being male individuals. The mean age was 59 years old in both groups. No significant difference of comorbidities was shown for benzbromarone users and allopurinol users (*p* > 0.05 for the Chi-square test). The proportions of the uses of thiazide diuretics, loop diuretics and aspirin were significantly different between benzbromarone users and allopurinol users (*p* < 0.001 for the Chi-square test). 

### 3.2. Incidence Density of the First Gout Flare

In Table 2, the benzbromarone group had a lower incidence rate of the first gout flare when compared with the allopurinol group (3.29 versus 5.46 per 1000 person-months, incidence rate ratio = 0.60, 95% CI = 0.56–0.64). As stratified by sex and age group, the incidence rate of the first gout flare was lower in the benzbromarone group than that in the allopurinol group. Individuals aged 20–39 in the allopurinol group showed the highest incidence rate of the first gout flare (7.01 per 1000 person-months). Males had a greater incidence rate of the first gout flare when compared with females no matter in the benzbromarone group or in the allopurinol group.

The Kaplan–Meier curve showed that the benzbromarone group had a lower cumulative incidence of the first gout flare as compared with the allopurinol group at the end of the cohort study (*p* < 0.0001, Figure 2).

### 3.3. The Hazard Ratio and 95% Confidence Interval of the First Gout Flare Associated with *Medications* and Comorbidities

After controlling for sex, age, thiazide diuretics, loop diuretics, aspirin, cerebrovascular disease, chronic kidney disease, coronary artery disease, DM, hyperlipidemia and hypertension, a multivariable Cox proportional hazards regression model showed that the adjusted HR of the first gout flare was 0.63 (95% CI = 0.59–0.68, *p* < 0.001) for the benzbromarone group when comparing with the allopurinol group (Table 3).

### 3.4. The Efficacy of Benzbromarone Use and Allopurinol Use

Table 4 presents the efficacy of benzbromarone and allopurinol at one local hospital in Taiwan. At 12 weeks of treatment, the serum level of uric acid was lower for the benzbromarone group than for the allopurinol group despite not achieving statistical significance (6.5 vs. 6.9 mg/dL, *p* = 0.492). The proportion of achieving treatment target at 12 weeks was higher in benzbromarone users than allopurinol users, despite not achieving a statistical significance (47.6% vs. 29.0%, *p* = 0.053). 

The cumulative defined daily dose of benzbromarone was higher in individuals who achieved treatment target than those who did not achieve the treatment target, with achieving statistical significance (mean ± standard deviation, 66.5 ± 25.2 vs. 58.8 ± 24.9, *p* = 0.029). There appeared to be a dose-dependent response for benzbromarone on the urate-lowering effect. The cumulative defined daily dose of allopurinol was higher in individuals who achieved treatment target than those who did not achieve treatment target, despite not achieving statistical significance (mean ± standard deviation, 46.0 ± 6.8 vs. 44.5 ± 13.0, *p* = 0.741).

## 4. Discussion

After adjusted for co-variables, our study showed that people with asymptomatic hyperuricemia who took benzbromarone had a lower hazard of developing the first gout flare when comparing with those who took allopurinol (HR = 0.63, Table 3). When compared with Rahlfs et al.’s study [22], the HR in our study was 0.63, which ranged from 0.567 to 0.789. The effect size seemed to be between small and medium. We consider our study to still be clinically relevant. 

Our hospital-based sub-analysis revealed that benzbromarone had a dose-dependent response on the effect of lowering uric acid. That is, the higher the dosage of benzbromarone, the higher proportion of achieving the treatment goal. After stratification by sex, our study showed that the incidence rate of the first gout flare was lower in the benzbromarone group than that in the allopurinol group no matter in males or in females. The incidence rate ratio was similar for males and females (Table 2), and thus there was no interaction between sex and medications on the risk of the first gout flare.

It would be impossible to adjust for all confounding factors; however, propensity score matching was employed to diminish the confounding effects. In our preliminary analysis, benzbromarone use and allopurinol use accounted for about 92% of the uric acid-lowering agent prescriptions in Taiwan. This is why we only included benzbromarone and allopurinol for analysis. It is difficult to measure the incidence rate of the first gout flare among users of probenecid and sulfinpyrazone owing to the small event number.

Some caveats need to be discussed. Clinically, hyperuricemia is classified into three types (overproduction of UA, renal underexcretion of UA and mixed type) [3]. Renal underexcretion of UA accounts for about 90% of people with hyperuricemia and the rest 10% are overproduction type [23,24]. Xanthine oxidase inhibitors are indicated for people with overproduction-type hyperuricemia [14,15]. Uricosuric agents are indicated for people with renal underexcretion-type hyperuricemia or for people with mixed-type hyperuricemia [14,15].

Four rules of prescription are discussed in order, including safety, efficacy, convenience and price. Benzbromarone, belonging to a uricosuric agent, is commonly used to manage people with renal underexcretion-type hyperuricemia [14,15]. Later, cases of benzbromarone-associated hepatotoxicity had been reported after 1994 [25,26]. Benzbromarone has been withdrawn from some European countries since 2003 [17]. Benzbromarone has been persistently prescribed in Taiwan until now; however, no serious side effects were reported even if there had been a high prevalence of hepatitis B carriers. 

Similarly, one population-based cohort study in South Korea showed that no serious side effects on liver were noted among benzbromarone users in 2002–2017 [27]. The real-world data from Taiwan and South Korea provide evidence that benzbromarone use is safe. Perhaps this is the reason why benzbromarone is the first urate-lowering agent of choice in Taiwan. Benzbromarone administration is easy, and it can be used orally once daily. The price of a 100 mg-benzbromarone tablet is about 0.17 US dollars in Taiwan, and thus the commercial profitability is low.

Allopurinol, belonging to a xanthine oxidase inhibitor (XO inhibitor), is mainly used to manage people with overproduction-type hyperuricemia [14,15]. Allopurinol potentially causes well-known cutaneous adverse reactions. Allopurinol ranked as the most-common one of ten suspicious drugs causing adverse drug reactions in Taiwan during 1999 to 2018 [28]. The incidence rate of cutaneous adverse reactions related to allopurinol therapy was 15.37 per 1000 person-years in Taiwan during 2005 to 2016 [29]. 

The incidence rate of hospitalization for allopurinol-associated cutaneous adverse events showed 0.69 per 1000 person-years in USA [30]. The incidence rate of cutaneous adverse events related to allopurinol therapy was higher in Taiwan when comparing with Korea and Japan [31]. The cutaneous adverse reactions found in Taiwan could be related to the HLA-B*58:01 genotype [32]. Allopurinol is used orally once daily. The price of a 100 mg-allopurinol tablet is 0.17 US dollars in Taiwan, and thus the commercial profitability is low.

Benzbromarone, probenecid and sulfinpyrazone are uricosuric agents. Benzbromarone has a better efficacy in lowering the serum UA followed by probenecid and sulfinpyrazone [33]. This can partially explain why benzbromarone was more frequently prescribed than probenecid and sulfinpyrazone in Taiwan. A head-to head prospective study revealed that only 53% of gout people taking allopurinol 300 mg/day (standard dosage) achieved the treatment goal (the serum UA level < 6 mg/dL); however, 100% of gout people taking benzbromarone 100 mg/day (standard dosage) achieved the treatment goal [34]. 

One cross-sectional study in Japan revealed that the proportion of achieving the treatment goal (the serum UA level ≤ 6 mg/dL) was higher in people with benzbromarone use than those with allopurinol use (70% versus 35.4%) [35]. One retrospective cohort study in the Netherlands revealed that the proportion of achieving the treatment goal (the serum UA level < 6 mg/dL) was greater in the benzbromarone group compared with the allopurinol group among men and women (76.1% versus 61% in men and 66.7% versus 58.4% in women, respectively) [36]. 

Our hospital-based sub-analysis revealed that 47.6% of benzbromarone users achieved the treatment target at 12 weeks but only 29.0% of allopurinol users achieved the treatment target at 12 weeks (serum uric acid < 6 mg/dL). All of these aforementioned findings support the concept that benzbromarone has a better efficacy on primary prevention of the first gout flare compared with allopurinol for asymptomatic hyperuricemia. We suggest that benzbromarone is superior to allopurinol in lowering the serum UA and in preventing the first gout flare for asymptomatic hyperuricemia.

Based on these profiles, including the predominant type of hyperuricemia being renal underexcretion, medication safety, therapeutic efficacy, oral administration once daily, as well as market price, it is not a rational decision to switch from benzbromarone to allopurinol. We suggest that if clinicians consider lowering the serum UA among people with asymptomatic hyperuricemia, benzbromarone should be regarded as the first drug of choice. We summarize the comparisons between benzbromarone use and allopurinol use in Table 5.

## 5. Limitation

Some limitations should be discussed. First, the data on the serum UA level were not available in the dataset. The cutoff point of hyperuricemia was not available in the dataset. We could not compare the serum UA levels at the baseline and at the first gout flare. Similarly, whether the serum goal of UA was achieved could not be assessed among those with the first gout flare. However, our hospital-based sub-analysis revealed that the use of benzbromarone had a better efficacy of achieving treatment target at 12 weeks when compared with the use of allopurinol. Owing to the small sample size, this did not achieve statistical significance. 

Second, according to the rule of the NHIP in Taiwan [37], febuxostat is only prescribed for people whose gout serum UA target cannot be achieved by taking other urate-lowering agents. That is, febuxostat is not indicated for people with asymptomatic hyperuricemia in Taiwan. That is why febuxostat was not included in our study. Third, one concern for the risk difference of the first gout flare might be due to the benzbromarone group being treated with an adequate dose, but the allopurinol group was treated with an inadequate dose. Based on the good quality of medical care in Taiwan, the treatment of hyperuricemia should be based on the standard dose. It is less likely that benzbromarone was used with an adequate dose but allopurinol was used with an inadequate dose. 

Fourth, theoretically we needed to select persons with asymptomatic hyperuricemia who had not used uric acid-lowering agents as a comparison group. However, the blood uric acid was not nationally examined in Taiwan. There were no records of the serum UA level among the national population. We were not able to divide the national population into two groups: hyperuricemia and normouricemia. Thus, the comparison group was difficult to be selected based on the claimed data. That is why we only compared patients on benzbromarone therapy with those on allopurinol therapy. 

We propose that only a randomized controlled trial has a chance to answer whether persons with asymptomatic hyperuricemia who take a uric acid-lowering agent could have a lower probability of developing the first gout flare as compared to those with asymptomatic hyperuricemia who do not use any uric acid-lowering agent because the baseline and surveillance conditions will be the same between the therapeutic group and the comparison group. Fifth, there appeared to be an imbalance in medication use between the benzbromarone group and the allopurinol group at baseline. At present, we did not have enough information to explain such a difference in detail.

## 6. Conclusions

People with asymptomatic hyperuricemia taking benzbromarone have a lower hazard of developing their first gout flare when compared with those taking allopurinol.

The use of benzbromarone achieved a higher proportion of the treatment goal compared with the use of allopurinol. There appears to be a dose-dependent response for benzbromarone on the urate-lowering effect. Based on the aforementioned advantages, we suggest that benzbromarone should be the first drug of choice if clinicians are treating asymptomatic hyperuricemia. 

## Figures and Tables

**Figure 1 jpm-12-00697-f001:**
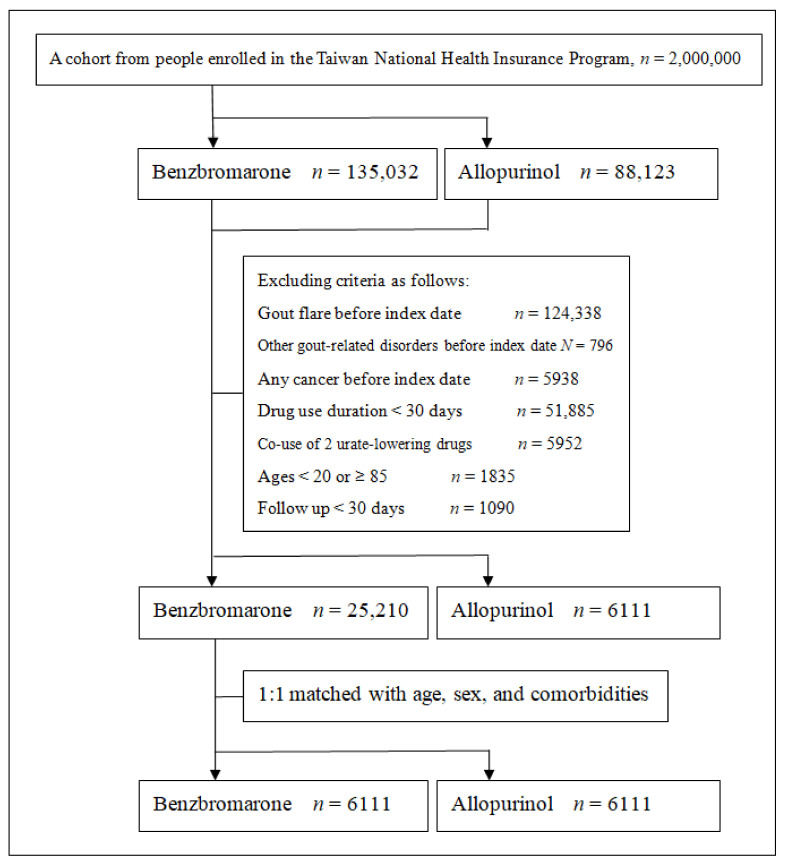
A flow chart of the selection of study subjects.

**Figure 2 jpm-12-00697-f002:**
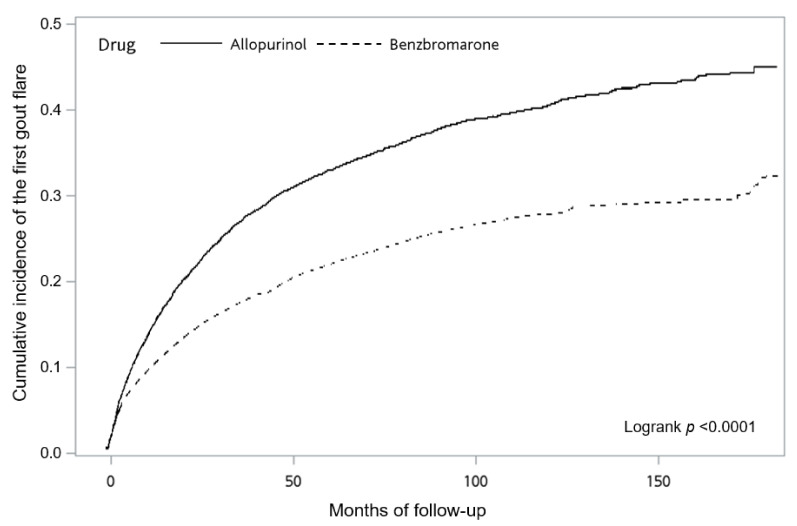
The Kaplan–Meier curve showed that the benzbromarone group had a lower cumulative incidence of the first gout flare than the allopurinol group at the end of the cohort study (*p* < 0.0001).

**Table 1 jpm-12-00697-t001:** Baseline information of the study subjects after propensity score matching.

	Benzbromarone *n* = 6111		Allopurinol *n* = 6111		
Variable	*n*	(%)	*n*	(%)	*p* Value *
Sex					1.00
Male	3989	(65.3)	3989	(65.3)	
Female	2122	(34.7)	2122	(34.7)	
Age (years)					0.310
20–39	646	(10.6)	683	(11.2)	
40–64	3001	(49.1)	3037	(49.7)	
65–84	2464	(40.3)	2391	(39.1)	
mean ± standard deviation ^†^	59.3 ± 14.6		58.9 ± 14.7		0.103
Baseline medications					
Ever use of thiazide diuretics	2120	(34.7)	2395	(39.2)	<0.001
Ever use of loop diuretics	3354	(54.9)	3876	(63.4)	<0.001
Ever use of aspirin	4085	(66.9)	4318	(70.7)	<0.001
Baseline comorbidities					
Cerebrovascular disease	622	(10.2)	682	(11.2)	0.078
Chronic kidney disease	1095	(17.9)	1094	(17.9)	0.981
Chronic obstructive pulmonary disease	518	(8.5)	568	(9.3)	0.112
Coronary artery disease	1108	(18.1)	1116	(18.3)	0.851
Diabetes mellitus	1891	(30.9)	1898	(31.1)	0.891
Hyperlipidemia	1663	(27.2)	1656	(27.1)	0.887
Hypertension	3631	(59.4)	3614	(59.1)	0.754

Data are presented as the number of subjects in each group, with the percentages given in parentheses. * Chi-square test; ^†^
*t*-test comparing the benzbromarone group with the allopurinol group.

**Table 2 jpm-12-00697-t002:** Incidence density of the first gout flare between the benzbromarone and allopurinol groups.

	Benzbromarone				Allopurinol				
Variable	*n*	Event	Person- Months	Incidence Rate	*n*	Event	Person- Months	Incidence Rate	Incidence Rate Ratio (95% CI) ^†^
All	6111	1433	435,840	3.29	6111	2273	416,195	5.46	0.60 (0.56–0.64)
Sex									
Male	3989	1048	271,139	3.87	3989	1691	255,168	6.63	0.58 (0.54–0.63)
Female	2122	385	164,701	2.34	2122	582	161,027	3.61	0.65 (0.57–0.74)
Age (years)									
20–39	646	159	50,363	3.16	683	322	45,929	7.01	0.45 (0.37–0.54)
40–64	3001	704	219,883	3.20	3037	1133	210,720	5.38	0.60 (0.54–0.65)
65–84	2464	570	165,594	3.44	2391	818	159,546	5.13	0.67 (0.60–0.75)

Incidence rate: 1000 person-months; ^†^ Incidence rate ratio: benzbromarone use versus allopurinol use (95% CI).

**Table 3 jpm-12-00697-t003:** Hazard ratio and 95% confidence interval of the first gout flare associated with medications and comorbidities.

	Crude			Adjusted ^†^		
Variable	HR	(95% CI)	*p* Value	HR	(95% CI)	*p* Value
Sex (male vs. female)	1.65	(1.54–1.78)	<0.001	1.68	(1.55–1.80)	<0.001
Age (every one year)	1.00	(0.99–1.00)	<0.001	1.00	(0.99–1.00)	<0.001
Benzbromarone use (allopurinol use as a reference)	0.62	(0.58–0.66)	<0.001	0.63	(0.59–0.68)	<0.001
Baseline medications						
Thiazide diuretics use (non-use as a reference)	1.10	(1.03–1.18)	0.003	1.19	(1.10–1.28)	<0.001
Loop diuretics use (non-use as a reference)	1.12	(1.05–1.20)	<0.001	1.22	(1.13–1.32)	<0.001
Aspirin use (non-use as a reference)	1.11	(1.03–1.19)	0.005	1.23	(1.14–1.33)	<0.001
Baseline comorbidities (yes vs. no)						
Cerebrovascular disease	0.88	(0.79–0.98)	0.024	0.88	(0.78–0.98)	0.021
Chronic kidney disease	0.80	(0.73–0.88)	<0.001	0.80	(0.73–0.88)	<0.001
Chronic obstructive pulmonary disease	0.96	(0.85–1.07)	0.452	-	-	
Coronary artery disease	0.90	(0.82–0.98)	0.015	0.86	(0.78–0.94)	0.001
Diabetes mellitus	0.76	(0.71–0.82)	<0.001	0.78	(0.72–0.84)	<0.001
Hyperlipidemia	0.84	(0.78–0.91)	<0.001	0.93	(0.86–1.01)	0.075
Hypertension	0.89	(0.84–0.95)	<0.001	0.97	(0.90–1.05)	0.494

^†^ Adjusted for significant variables found in a crude analysis including for sex, age, thiazide diuretics use, loop diuretics use, aspirin use, cerebrovascular disease, chronic kidney disease, coronary artery disease, diabetes mellitus, hyperlipidemia and hypertension.

**Table 4 jpm-12-00697-t004:** The serum uric acid and the proportion of achieving treatment target between benzbromarone users and allopurinol users at one local hospital in Taiwan.

	Benzbromarone *n* = 205	Allopurinol *n* = 31	
Variable	*n* (%)	*n* (%)	*p* Value *
Sex (male)	177 (85.9)	25 (80.7)	0.440
Age (years) mean ± standard deviation ^†^	56.3 ± 15.4	57.6 ± 12.1	0.658
Baseline uric acid (mg/dL)			
mean ± standard deviation ^†^	9.1 ± 1.9	9.5 ± 2.0	0.406
Serum uric acid at 12 weeks treatment (mg/dL)	6.5 ± 2.4	6.9 ± 1.9	0.492
mean ± standard deviation ^†^			
Achieving target at 12 weeks treatment ^#^	98(47.6)	9(29.0)	0.053

* Chi-square test; ^†^
*t*-test comparing the benzbromarone and allopurinol groups. ^#^ Achieving target: serum uric acid < 6 mg/dL.

**Table 5 jpm-12-00697-t005:** Comparison between benzbromarone use and allopurinol use.

	Benzbromarone	Allopurinol
Hyperuricemia type	underexcretion type (predominant type)	overproduction type
Medication safety	superior (hepatotoxicity not seen in Taiwan)	cutaneous adverse events (more common in Taiwan when comparing with Korea and Japan)
Lowering serum uric acid	superior	
Reducing the risk of the first gout flare	superior (found in our study)	
Use convenience	oral use and once daily	oral use and once daily
Market price	similar	similar

## Data Availability

The original contributions presented in the study are included in the article. Further inquiries can be directed to the first author.

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
