# Peer review of "Comparison of Benzbromarone and Allopurinol on Primary Prevention of the First Gout Flare in Asymptomatic Hyperuricemia"

_jpm, 2022, doi:10.3390/jpm12050697_

Round 1

Reviewer 1 Report

The authors provide a retrospective study regarding the primary prevention of a first gout flare in patients with asymptomatic hyperuricemia treated with allopurinol or benzbromarone using data from the National Health Insurance in Taiwan. In addition, they performed a sub analysis of the effectiveness of both drugs regarding the reduction of uric acid levels in blood.

The overall message of this analysis is clinically relevant and reinforces previously published data, supporting a higher efficacy of benzbromarone compared to allopurinol in reducing gout flares in this group of patients, as well as a tendency towards higher rates of UA normalization after 12 weeks of treatment.

Some of the strengths of the analysis include the large sample, the use of a propensity score matching in order to equalize both groups of comparison and the clarity in pointing out some of the limitations of the study. In addition, the authors considered possible confounders, such as the use of diuretics in a multivariable analysis (however, it was not presented in the sub-analysis)

The paper has clinical relevance, once it helps guiding decision making process in patients in which the treatment of hyperuricemia may be indicated.

However, there are still some important concerns that should be addressed, in order to reduce residual doubt and clarify some of the results: 

  • Even though the frequency of first gout flare is described for both drugs, it was not possible to include a control group. This would be of extreme importance, since one of the main goals of the analysis was to evaluate the efficacy of those drugs in the PRIMARY PREVENTION of first gout flare in these individuals. It would be fundamental to know how both drugs perform compared to the standard of care (no pharmacological intervention), especially in order to understand how big the effect size is. The authors found a statistically significant difference favoring benzbromarone. Is that also clinically relevant? How big is the effect size? Please address this issue in the discussion.
  • The authors used T-tests to compare continuous variables. Where these tested for normality? If so, please describe.
  • Was the treatment compliance also considered in this analysis? This is especially relevant for those patients who presented gout flares. How long did these patients take the drugs? Was the treatment interrupted once normal UA levels were achieved? What was the average time of follow up for patients on both groups? Understanding the patterns of use of the drug is very important in order to interpret the results of the comparison. In addition, as the authors describe, that allopurinol can lead to allergic reactions of the skin. Could that have influenced compliance or dosing?
  • Unfortunately, the mean/median dose of each drug could not be investigated, which could also impact the results. The authors discuss that on paragraph 31, however phrasing is not clear. Please readdress this issue.
  • The authors discuss that allopurinol is mainly used for treating patients with overproduction type hyperuricemia, whereas benzbromarone for those with renal underexcretion. What is the exact indication for prescribing one or the other drug according to the local authorities? Is it possible that patients with “more difficult to treat” hyperuricemia received allopurinol?
  • In the sub analysis there is no mention to other possible confounders, such as CKD. Hypertension, diabetes and use of ASS and diuretics (the latter, was shown to be statistically higher in the allopurinol group in the first analysis).
  • What was the incidence of gout in the patients treated with benzbromarone that were not included in the matching? It is a large sample, results could be interesting.

Author Response

Reviewer 1 

Comments and Suggestions for Authors

The authors provide a retrospective study regarding the primary prevention of a first gout flare in patients with asymptomatic hyperuricemia treated with allopurinol or benzbromarone using data from the National Health Insurance in Taiwan. In addition, they performed a sub analysis of the effectiveness of both drugs regarding the reduction of uric acid levels in blood.

The overall message of this analysis is clinically relevant and reinforces previously published data, supporting a higher efficacy of benzbromarone compared to allopurinol in reducing gout flares in this group of patients, as well as a tendency towards higher rates of UA normalization after 12 weeks of treatment.

Some of the strengths of the analysis include the large sample, the use of a propensity score matching in order to equalize both groups of comparison and the clarity in pointing out some of the limitations of the study. In addition, the authors considered possible confounders, such as the use of diuretics in a multivariable analysis (however, it was not presented in the sub-analysis)

The paper has clinical relevance, once it helps guiding decision making process in patients in which the treatment of hyperuricemia may be indicated.

However, there are still some important concerns that should be addressed, in order to reduce residual doubt and clarify some of the results: 

ï‚·        Even though the frequency of the first gout flare is described for both drugs, it was not possible to include a control group. This would be of extreme importance, since one of the main goals of the analysis was to evaluate the efficacy of those drugs in the PRIMARY PREVENTION of first gout flare in these individuals. It would be fundamental to know how both drugs perform compared to the standard of care (no pharmacological intervention), especially in order to understand how big the effect size is. The authors found a statistically significant difference favoring benzbromarone. Is that also clinically relevant? How big is the effect size? Please address this issue in the discussion.

Re Response:

Rahlfs V, Zimmermann H. Effect size measures and their benchmark values for quantifying benefit or risk of medicinal products. Biom J. 2019;61(4):973-982.

When compared with Rahlfs et al’s study, the HR in our study was 0.63, which ranged from 0.567 to 0.789. The effect size seemed to be between small and medium. We think that our study is still clinically relevant.  Please see below. This point is added in discussion section. Thank you for good comments.

ï‚·        The authors used T-tests to compare continuous variables. Where these tested for normality? If so, please describe.

Response: Altman DG, Bland JM. Statistics notes: the normal distribution. Bmj. 1995;310(6975):298. We review this paper.
According to the central limit theory of Altman et al’s study, if the sample population is approximately normal, then the sampling distribution will also be normal. Our sample size was large enough, so the violation of the normality assumption is less likely to cause major problems. Thank you for good comments.

ï‚·        Was the treatment compliance also considered in this analysis? This is especially relevant for those patients who presented gout flares. How long did these patients take the drugs? Was the treatment interrupted once normal UA levels were achieved? What was the average time of follow up for patients on both groups? Understanding the patterns of use of the drug is very important in order to interpret the results of the comparison. In addition, as the authors describe, that allopurinol can lead to allergic reactions of the skin. Could that have influenced compliance or dosing?

Response:

1.     This study was a retrospective study based claims data. The treatment compliance could not be determined in our study. We are not sure whether the treatment was interrupted once normal UA levels were achieved. Similarly, we are not sure whether allergic reactions of allopurinol could have influenced compliance. We think that only a randomized controlled trial has a chance to answer whether persons with asymptomatic hyperuricemia who take a uric acid-lowering agent could have a lower probability of developing the first gout flare as compared to those persons with asymptomatic hyperuricemia who do not use any uric acid-lowering agent because the baseline and surveillance conditions will be the same between the therapeutic group and the comparison group. This point is added in limitation section. Thank you for good comments.

2.     About 92% of benzbromarone users took benzbromarone ≦ 540 days and 8% took benzbromarone > 540 days. About 90% of allopurinol users took allopurinol ≦ 540 days and 10% took allopurinol > 540 days.

ï‚·        Unfortunately, the mean/median dose of each drug could not be investigated, which could also impact the results. The authors discuss that on paragraph 31, however phrasing is not clear. Please readdress this issue.

 Response: This point is readdressed. Our hospital-based sub-analysis revealed that 47.6% of benzbromarone users would achieve the treatment target at 12 weeks but only 29.0% of allopurinol users would achieve the treatment target at 12 weeks (serum uric acid < 6 mg/dl). Thank you for good comments.

ï‚·        The authors discuss that allopurinol is mainly used for treating patients with overproduction type hyperuricemia, whereas benzbromarone for those with renal underexcretion. What is the exact indication for prescribing one or the other drug according to the local authorities? Is it possible that patients with “more difficult to treat” hyperuricemia received allopurinol?

Response: Renal underexcretion of UA accounts for about 90% of people with hyperuricemia and the rest 10% are overproduction type. It is reasonable to prescribe benzbromarone first clinically. In our preliminary analysis, benzbromarone and allopurinol accounted for about 92% of urate-lowering agent prescriptions in Taiwan. This point is added in discussion section. Thank you for good comments.

ï‚·        In the sub-analysis there is no mention to other possible confounders, such as CKD. Hypertension, diabetes and use of ASS and diuretics (the latter, was shown to be statistically higher in the allopurinol group in the first analysis).

Response: The sub-analysis was a preliminary analysis to support our main study. The sample size was small, so the confounders could not be controlled. Your good comments indicate a future research direction. Thank you for good comments.

ï‚·        What was the incidence of gout in the patients treated with benzbromarone that were not included in the matching? It is a large sample, so the results could be interesting.

 Response: The original data about patients treated with benzbromarone that were not included in the matching have been deleted from our computer. We cannot provide the results. Sorry!

All changes were underlined in blue.

Thanks for your very helpful comments.

Reviewer 2 Report

The authors sought to compare the treatment outcomes of benzbromarone vs allopurinol in a group of patients with asymptomatic hyperuricemia for a first episode of gout in a national administrative database. It is an important work in the sense of allowing the use of treatments that avoid the first episode of gout that radically changes the natural history of hyperuricemia, and methodologically it is well thought out (including propensity score in a heterogeneous group of individuals).

It is important to mention that real-world evidence studies are characterized by having a structuring of the information intended to be measured to substantially reduce the potential biases attributed to this methodology.

The main problem with this work is that the use of benzbromarone is indicated only in patients who do not respond to or do not tolerate treatment with allopurinol; in this sense, using the molecule as prophylaxis to prevent gout attacks in hyperuricemia would be outside the context of its current approval. Its use is very restricted due to its high hepatotoxicity, in some cases fatal.

The authors speak of the prices of the molecules and the incidence of adverse events of each molecule; in this section it would be highly desirable to have a comparison of the cost-effectiveness of the molecules according to these outcomes.
Finally, the discussion should clearly include a section mentioning the restrictions on its use and the scope of the results of this study, in terms of countries (or national health systems) where it is possible to use benzbromarone, since its "prophylactic" use due to its potential risk of hepatotoxicity limits its use as a public health policy in the population with hyperuricemia.

Author Response

 Reviewer 2

Comments and Suggestions for Authors

The authors sought to compare the treatment outcomes of benzbromarone vs allopurinol in a group of patients with asymptomatic hyperuricemia for a first episode of gout in a national administrative database. It is an important work in the sense of allowing the use of treatments that avoid the first episode of gout that radically changes the natural history of hyperuricemia, and methodologically it is well thought out (including propensity score in a heterogeneous group of individuals).

It is important to mention that real-world evidence studies are characterized by having a structuring of the information intended to be measured to substantially reduce the potential biases attributed to this methodology.

The main problem with this work is that the use of benzbromarone is indicated only in patients who do not respond to or do not tolerate treatment with allopurinol; in this sense, using the molecule as prophylaxis to prevent gout attacks in hyperuricemia would be outside the context of its current approval. Its use is very restricted due to its high hepatotoxicity, in some cases fatal.

Response: Renal underexcretion of UA accounts for about 90% of people with hyperuricemia and the rest 10% are overproduction type. It is reasonable to prescribe benzbromarone first clinically. In our preliminary analysis, benzbromarone and allopurinol accounted for about 92% of urate-lowering agent prescriptions in Taiwan. This point is added in discussion section.

Theoretically, we should select people with asymptomatic hyperuricemia who did not take urate-lowering agents as the comparison group. However, serum uric acid is not routinely examined in Taiwan. It is hard to determine who has asymptomatic hyperuricemia. It would be difficult to compare the incidence rate of the first gout attack between the therapeutic group and the comparison group by using a claim database. We think that only a randomized controlled trial has a chance to answer whether persons with asymptomatic hyperuricemia who take a uric acid-lowering agent could have a lower probability of developing the first gout flare as compared to those persons with asymptomatic hyperuricemia who do not use any uric acid-lowering agent because the baseline and surveillance conditions will be the same between the therapeutic group and the comparison group.  This point is added in limitation section. Thank you for good comments.

The authors speak of the prices of the molecules and the incidence of adverse events of each molecule; in this section it would be highly desirable to have a comparison of the cost-effectiveness of the molecules according to these outcomes.

Response: The price of a 100mg-benzbromarone tablet is about 0.17 US dollars in Taiwan and the price of a 100mg-allopurinol tablet is also 0.17 US dollars in Taiwan. The commercial profitability of both drugs is low.  Benzbromarone has been persistently prescribed in Taiwan till now, but no serious side effects were reported even if there had been a high prevalence of hepatitis B carriers. The incidence rate of cutaneous adverse reactions related to allopurinol therapy was 15.37 per 1,000 person-years in Taiwan during 2005 to 2016. We make an assumption that benzbromarone has a better cost-effectiveness than allopurinol. We summarize the comparisons between benzbromarone use and allopurinol use in Table 5. Please see below. Thank you for good comments.

Finally, the discussion should clearly include a section mentioning the restrictions on its use and the scope of the results of this study, in terms of countries (or national health systems) where it is possible to use benzbromarone, since its "prophylactic" use due to its potential risk of hepatotoxicity limits its use as a public health policy in the population with hyperuricemia.

Response: We summarize the comparisons between benzbromarone use and allopurinol use in Table 5, including hyperuricemia type, medication safety, therapeutic efficacy, use convenience, as well as market price. Thank you for good comments.

Table 5. Comparison between benzbromarone use and allopurinol use

Benzbromarone

Allopurinol

Hyperuricemia type

underexcretion type

(predominant type)

overproduction type

Medication safety

superior (hepatotoxicity not seen in Taiwan)

cutaneous adverse events (more common in Taiwan when comparing with Korea and Japan)

Lowering serum uric acid

superior

Reducing the risk of the first gout flare

superior

(found in our study)

Use convenience

oral use and once daily

oral use and once daily

Market price

similar

similar

 All changes were underlined in blue.

Thanks for your very helpful comments.

Round 2

Reviewer 1 Report

The authors were able to answer the questions sufficiently. The study limitations are now clear in the manuscript.

Reviewer 2 Report

The authors made the appropriate corrections and I agree with the publication.